# Chloride Applied via Fertilizer Affects Plant Nutrition and Coffee Quality

**DOI:** 10.3390/plants12040885

**Published:** 2023-02-15

**Authors:** César Santos, Marcelo Ribeiro Malta, Mariana Gabriele Marcolino Gonçalves, Flávio Meira Borém, Adélia Aziz Alexandre Pozza, Herminia Emilia Prieto Martinez, Taylor Lima de Souza, Wantuir Filipe Teixeira Chagas, Maria Elisa Araújo de Melo, Damiany Pádua Oliveira, Alan Dhan Costa Lima, Lívia Botelho de Abreu, Thiago Henrique Pereira Reis, Thaís Regina de Souza, Victor Ramirez Builes, Douglas Guelfi

**Affiliations:** 1Department of Soil Science, Federal University of Lavras, Lavras 37203-202, Brazildouglasguelfi@ufla. br (D.G); 2Agricultural Research Company of Minas Gerais (EPAMIG), Belo Horizonte 31170-495, Brazil; 3Department of Agricultural Engineering, Federal University of Lavras, Lavras 37203-202, Brazil; 4Department of Agronomy, Federal University of Viçosa, Viçosa 36570-900, Brazil; 5Center for Plant Nutrition and Environmental Research Hanninghof, Yara International, 48249 Dülmen, Germany

**Keywords:** blend fertilizers, chlorine, cup test, polyphenol oxidase

## Abstract

The present study had the objective to evaluate the effect of blends of KCl and K_2_SO_4_ fertilizers and their influence on the yield and the nutritional state of coffee plants, as well as on the chemical composition and quality of the coffee beverage. The experimental design was in randomized blocks with four repetitions and six treatments (T1: 100% KCl; T2: 75% KCl + 25% K_2_SO_4_; T3: 50% KCl + 50% K_2_SO_4_; T4: 25% KCl + 75% K_2_SO_4_; T5: 100% K_2_SO_4_; and a control, without application of K). The following analyses were performed: K and Cl content in the leaves and the soil, stocks of Cl in soil, yield, removal of K and Cl with the beans, cup quality of the beverage, polyphenol oxidase activity (PPO), electric conductivity (EC), potassium leaching (KL), the content of phenolic compounds, the content of total sugars (TS), and total titratable acidity (TTA). The stocks of Cl in the soil decreased as the proportion of KCl in the fertilizer was reduced. The fertilization with KCl reduces the cup quality and the activity of the polyphenol oxidase, probably due to the ion Cl. The increase in the application of Cl directly relates to the increase in potassium leaching, electric conductivity, and titratable acidity. Indirectly, these variables indicate damages to the cells by the use of Cl in the fertilizer. The activity of the polyphenol oxidase enzyme and the cup quality indicate that the ion Cl- reduces the quality of the coffee beverage. K content in the leaves was not influenced by the application of blends of K fertilizer while Cl content increased linearly with KCl applied. The application of KCl and K_2_SO_4_ blends influenced coffee yield and the optimum proportion was 25% of KCl and 75% of K_2_SO_4_. The highest score in the cup quality test was observed with 100% K_2_SO_4_.

## 1. Introduction

Coffee is one of the most popular beverages in the world, and its cultivation is widespread in 80 countries. Brazil is the largest producer and second largest consumer of coffee in the world. The gross revenue of “Cafés do Brasil” in the 2022 harvest was R$61.82 billion with *Coffea arabica* accounting for 77% the total revenue (R$ 47.48 billion). The state of Minas Gerais is the largest producer, responding for R$ 33.28 billion or 54% of national revenues [1]. After petroleum, coffee is the second most commercialized product [2,3]. The marketing price is based on the quality of the beverage, which is related to the physical, chemical, and sensorial characteristics of the product [2,4,5].

Fertilization and crop nutrition can influence both yield and the chemical composition of the raw beans, which, consequently, interfere with the quality of the beverage [6]. After nitrogen, potassium (K) is the most accumulated nutrient in coffee plant fruits, where it is demanded in high amounts. K is related to the enzymatic activation of several metabolic processes, such as photosynthesis, proteins, and carbohydrates synthesis, and in the maintenance of cell turgidity [7,8,9]. In addition, K is directly related to the transport of sugars from the source to the drain (fruits) [10].

The effects of the accompanying chloride ion (Cl) of the potassium chloride fertilizer (KCl) are currently under debate. Cl is demanded by the plants at low amounts, thus being one of the last micronutrients to enter the micronutrient list. Its role is related to the water photolysis on the photosystem II, enzyme activation (amylase, asparagine synthetase, and tonoplast ATPase), and stomatal control [11].

Despite Cl being essential to plant nutrition, when accompanying a highly demanded macronutrient such as K, it can reach excessive concentrations in the soil and plants [12] and consequently reduce the quality of the beverage. In coffee plants, high concentrations of Cl are related to the increase in plant water, which favors an undesirable fermentation of the fruits by microorganisms [13,14].

The study of the Cl influence on the quality of the coffee beverage is not recent, but it is still inconclusive. For example, Dias et al. [15] evaluated an alternative source of K (glauconite silicate mineral) to KCl in coffee fertilization. Despite finding similar yields and polyphenol oxidase activity (PPO) in the beans, the use of the glauconite did not improve the sensorial quality. Silva et al. [16], along two seasons, verified that fertilization with potassium sulfate (K_2_SO_4_) increased PPO activity in comparison with KCl, which, according to the authors, is indicative of a better beverage quality. These studies suggest possible negative effects of the Cl on the quality of the coffee beverage and the necessity to use K sources without Cl as the accompanying ion, such as K_2_SO_4_ (48% K_2_O, 16% S) and potassium nitrate (44% K_2_O, 13% N). Nonetheless, this could increase the production costs, as these sources are more expensive than KCl. In turn, blends of KCl and K_2_SO_4_ (a physical mixture of the two less expensive sources in the market) could be an alternative to reduce Cl to thresholds that do not affect the quality of the coffee beverage without excessively increasing the costs of the fertilization.

Therefore, the present study had the objective to evaluate the effect of blends of KCl and K_2_SO_4_ fertilizers at different proportions and their influence on the yield and the nutritional state of coffee plants, as well as on the chemical composition and quality of the coffee beverage.

## 2. Results

### 2.1. Effects of the KCl and K_2_SO_4_ Blends in the Stocks of Cl in the Soil, Nutrition, and Yield of Coffee Plants

#### 2.1.1. Harvest of 2017/2018

The initial content of K in the 0–20 and 20–80 cm layers was 91.5 and 58.6 mg dm^−3^ while stocks of the element were 201.3 and 386.7 kg ha^−1^, respectively. The content of Cl in the 0–20 and 20–80 cm layers was 153.8 and 203.8 mg dm^−3^ while stocks were 338.3 and 1345.0 kg ha^−1^, respectively. K and Cl contents in the leaves were 19.2 g kg^−1^ and 2880 mg kg^−1^, respectively (Table 1).

Soil stocks of Cl were influenced by the K blends (Appendix A). Overall, the amount of Cl decreased along with the KCl proportion in the treatment. In the 0–20 cm layer, Cl stocks with T1, T2, and T3 were similar (~190 kg ha^−1^). In the 20–80 cm layer, the highest Cl stock was in the T1 (622 kg ha^−1^), T2 (513 kg ha^−1^), and T3 (409 kg ha^−1^; which did not differ from T2). Other treatments showed similar Cl stocks.

In this harvest, K content varied from 20.8 to 34.8 g kg^−1^ (Appendix A). Cl content decreased with the increase in K_2_SO_4_, with results ranging from 3644 to 5275 mg kg^−1^. For the statistically non-significant variables, mean values for Cl content in the beans (Appendix A), yield (Appendix A), and Cl removal (Appendix A) were 1778 mg kg^−1^, 3631 kg ha^−1^ (Appendix A), and 2.8 kg ha^−1^, respectively. The lowest K removal by the beans was in T3 (11.3 kg ha^−1^). The K removal from other treatments had means close to 24 kg ha^−1^ (Appendix A).

#### 2.1.2. Harvest of 2018/2019

In this harvest, treatments showed Cl stocks near 65 kg ha^−1^ in the 0–20 cm layer (Figure 1). The lowest Cl amount was stocked with T4 (50 kg ha^−1^). In the 20–80 cm layer, the average stock of the six treatments was 122 kg ha^−1^.

K content in the leaves varied from 17 to 21 g kg^−1^, and the lowest value was in the control (Figure 2). Overall, Cl content in the leaves decreased with less KCl applied. Treatments T1 (6950 mg kg^−1^) and T2 (7621 mg kg^−1^) were far superior from the others.

There was no significant differences among treatments for the following variables: Cl content in beans (693 mg kg^−1^; Figure 3A), yield (1804 kg ha^−1^; Figure 3A), and K removal (6.0 kg ha^−1^) and Cl removal (0.21 kg ha^−1^; Figure 3B).

#### 2.1.3. Harvest of 2019/2020

The same pattern of the first harvest was observed. Cl stocks in the soil and Cl content in the leaves decreased along with the proportion of KCl in the blend (Figure 4 and Figure 5). In the 0–20 cm layer, Cl stocks were higher for T1 (119 kg ha^−1^) and T3 (101 kg ha^−1^) and lower for T5 (57 kg ha^−1^) and the control (54 kg ha^−1^); in the 20–80 cm layer, the lowest values occurred with T5 and control (~267 kg ha^−1^). The highest Cl content in the leaves was in T1 (4919 mg kg^−1^), and the lowest content was found in T5 (1762 mg kg^−1^) and control (1819 mg kg^−1^).

There were differences in the yield of the coffee plants depending on the treatment, although they all received the same dose of K (Figure 6A). Yields of T1 (4147 kg ha^−1^) and the control (4055 kg ha^−1^) were the lowest. The treatments that received K_2_SO_4_ up to 75% of applied K had similar yields (~5100 kg ha^−1^). Yields of T2, T3, and T4 were 19, 24, and 24% higher than the yield of T1. Treatment T5 yield was similar to the best yields but did not differ from the yield of T1.

There was no significant difference among treatments for the other agronomic variables. The average results were as follows: 883 mg kg^−1^ for the content of Cl in the beans (Figure 6A), 45 kg ha^−1^ for the K removal, and 1.6 kg ha^−1^ for the Cl removal by the beans (Figure 6B).

### 2.2. Effect of KCl and K_2_SO_4_ Blends on the Chemical Composition and Quality of the Coffee Beverage

#### 2.2.1. Harvest of 2017/2018

The highest K leaching (KL) was in T3 (36.7 µg g^−1^), and the lowest was in T1 (30 µg g^−1^) (Appendix A). In the variables related to the quality of the coffee beverage, there was a similar pattern among the treatments despite not having differences from each other. The resulting averages were: 81 points for the sensorial analysis; 91.8 µS cm^−1^ g^−1^ for the electric conductivity (EC); 9.7% for total sugars (TS); 1.04% for caffeine content (Caf); 47.6 u min^−1^ g^−1^ for the activity of polyphenol oxidase (PPO); 186.5 mL NaOH 100 g^−1^ of sample for total titratable acidity (TTA); and 6.4% for polyphenols (Pol) (Appendix A).

#### 2.2.2. Harvest of 2018/2019

All treatments received over 80 points in the cup quality (Figure 7A). The highest scores were achieved by treatments T3 (83 points), T4 (84.5), and T5 (83).

The EC was higher in T1 (221 µS cm^−1^ g^−1^) and lower in T5 (132 µS cm^−1^ g^−1^) and the control (117 µS cm^−1^ g^−1^). T2 was 24% lower than T1 (Figure 7B). Potassium leached (KL) more in treatments where the proportion of KCl was higher than K_2_SO_4_ (Figure 7B). T1, T2, and T3 had similar KL (~37 µg g^−1^) while the other treatments were lower (~29 µg g^−1^). There was no significant variation for the other variables, and the means were: 9.1% for TS, 1.03% for Caf, 46 u min^−1^ g^−1^ for PPO, 190 mL NaOH 100 g^−1^ of sample for TTA, and 5.0% for the content of Pol (Table 2).

#### 2.2.3. Harvest of 2019/2020

The highest grade was achieved in T5 (89 points) and the lowest was in the control (84) (Figure 8). However, T1 was not different from the control. T3 and T4 had similar scores (86 points) while T2 was similar to T3 and T4, but not different from T1. The other variables were not influenced by the application of the blends of K. The following means were found: 124 µS cm^−1^ g^−1^ for EC, 9.6% for TS, 1.02% for Caf, 54 u min^−1^ g^−1^ for PPO, 70.9 µg g^−1^ for KL, 195 mL NaOH 100 g^−1^ of sample for TTA, and 5.0% for Pol (Table 3).

#### 2.2.4. Principal Component Analysis (PCA) for the Agronomic Variables, Chemical Composition of the Beans, and Quality of the Coffee Beverage

The PCAs allow one to understand the behavior of the variables related to the chemical composition and quality of the coffee beverage in relation to the treatments, even if some of the variables were not statistically significant. Initially, we attempted to separate the treatments with ellipses in the PCAs calculated with all available data. However, we decided to add these PCAs in the Appendix A as we observed a lower percentage of the explained variance in comparison with the PCAs calculated without the agronomic data (those related to the soil). This happens when a set of variables with very different origins (soil, beverage quality, leaf composition) are used in PCA calculations (S5, S6, and S7).

Therefore, the treatments in the following PCAs represent the proportions of KCl and K_2_SO_4_ as described before. Thus, the closer they are, the greater the correlation between the variables that constitute these treatment groups. The agronomic variables stocks of K and Cl in the 0–20 and 20–80 cm layers, K and Cl contents in the leaves, yield, K and Cl removal in the beans and Cl content in the beans are supplementary variables; that is, they do not contribute to explaining the variability of the data. These illustrative variables are represented as dashed arrows, and they help to interpret the other data.

The PCA for the first harvest (2017/2018) indicates that the two components (Dim1 and Dim2) responded for 51.7% of the total variability of the data. The variances explained by these two variables were 38.2% and 18.9%, respectively (Appendix A). The variables K and Cl removal by the beans, K content in the leaves, yield, TS, Pol, and Caf were highly correlated. The control treatment was more related to these variables. In addition, these variables were negatively correlated to the stock of K in the 20–80 cm layer, stock of Cl in the 0–20 cm layer, and cup quality. The EC, KL, and TTA variables were highly correlated. The variables Cl content in the beans, stock of K in the 0–20 cm layer, and Cl content in the leaves were lowly correlated with the stock of Cl in the 20–80 cm layer and with the PPO activity. Furthermore, these variables were negatively correlated with the variables EC, KL, and TTA.

In the harvest of 2018/2019, the two PCA components explained 54.9% of the variability. The variances of each component were 39.6% and 15.3%, respectively (Figure 9). The cup quality was positively correlated with TS in the coffee beans and with yield. These variables are also correlated with T2. The KL variable was correlated with the stocks of Cl in both layers and with the Cl content in the leaves of the plants. T1 and T4 were close to these variables. Pol, Caf, and K and Cl in the beans were strongly correlated. Overall, the PCA shows that the quality of the coffee beverage is negatively correlated with the content of Cl in the leaves and beans.

In the last harvest (2019/2020), the PCA explained 56.8% of the variability with the first component explaining 36.8%, and the second component explained 20% of the variance (Figure 10). The cup quality and PPO activity were closely related to T5. The stock of Cl in the 20–80 cm layer was strongly related to T1. KL and Cl contents in the leaves were also correlated with T1.

## 3. Discussion

Despite the long-time fertilization with KCl in the area, the initial K stocks in the soil were considered medium level [17]. For the initial amount of Cl, however, there is no method of extraction and no reference values to relate to the needs of coffee plants. Cl is a micronutrient that is required in low amounts by plants. Under field conditions, Cl deficiency is uncommon while the excess is frequently expressed.

The stocks of Cl reduced during the three years of study due to the leaching of the element to deeper layers in the soil. The Cl ion has low interaction with the soil solid phase [18]; thus, it is easily leachable [12].

There was a tendency to accumulate K in the leaves when plants received more KCl. KCl fertilizer is more soluble than K_2_SO_4_. Nonetheless, in all harvests, the foliar content of K remained adequate in the range of 19.7 to 31 g kg^−1^ [19,20] except for the low content in the control treatment in the last two harvests.

The Cl content in the leaves in all harvests was reduced from T1 to T5 and the control. The content of Cl usually found in plant tissues ranges from 2000 to 30,000 mg kg^−1^, which is equivalent to the amount of macronutrients [21,22]. However, plants vary in their tolerance to Cl [23]. According to Marschner [22], plants sensitive to Cl show toxicity symptoms at concentrations higher than 3500 mg kg^−1^. In tolerant plants, the symptoms appear when the concentration range from 20,000 to 30,000 mg kg^−1^.

Under field conditions, toxicity symptoms caused by Cl excess are uncommon. Symptoms are characterized by the reduction of the width of the leaves, with possible curling, and the presence of wide necrosis with later leaf drying [11,12]. In this study, despite the high content found when KCl was applied (over 2500 mg kg^−1^), plants did not show toxicity symptoms. However, it is important to emphasize the damages to the metabolism, growth, and yield that can occur even in concentrations below the toxicity threshold. In fact, in the harvest of 2019/2020, when the foliar content of Cl reached 4919 mg kg^−1^ in treatment T1, a lower yield was observed. An argument could be made for the higher availability of S in the treatments that received more K_2_SO_4_, but despite the source of K, all treatments received 2 t ha^−1^ of gypsum, which provided 340 kg ha^−1^ of S to the soil. The reduction in the yield is probably more related to the excess of Cl than the lack of S in the fertilization. Another aspect to be taken into account is that, under high concentrations of Cl in the soil, anion–anion competition may occur mainly with phosphate and nitrate ions. This is due to the inability of proteins to differentiate among nitrate, phosphate, and chlorine ions, leading to the absorption of the ion in higher concentration [12].

Conversely, the reduction in the yield is notable, even with no clear statistical separation, between the treatment that received only K_2_SO_4_ and the treatment fertilized solely with KCl. It is possible that such difference may be related to the solubility of the K_2_SO_4_, 80 g L^−1^ at 25 °C, which is considerably lower than the solubility of the KCl, 279 g L^−1^ [24]. This difference in the availability of K can compromise the yield in harvests of increased productivity as the harvest of 2019/2020. Another explanation is that the high solubility of KCl can benefit the absorption of cations, such as K, Ca, and Mg [25], increasing plant nutrition, even though it is for a short period. A third possibility is the excess of SO_4_^2−^, limiting the availability and absorption of H_2_PO_4_^−^ by the plant, since, besides the K_2_SO_4_ application, gypsum was also applied.

The results suggest advantages in providing the two sources of K (25 to 75% of K_2_SO_4_) to increase yield. In the first two years of the experiment, when the yields were lower, the exportation of K by the beans was less intense and plant production was not limited by the sources of K once they were applied at the same dose.

The removal of K and Cl and the content of Cl in the beans were not different among the treatments since these elements remain in high concentration in the mucilage and the bean peels [26]. The exception is treatment T3 at the first harvest, but that might be related more to the history of the area than to the treatments.

### 3.1. Effects of the Application of KCl and K_2_SO_4_ Blends in the Chemical Composition of the Beans and in the Quality of the Coffee Beverage

There was a response in the KL in the first and second harvests. This variable is related to the integrity of the cell wall and membrane and, consequently, to the coffee beverage quality. When these structures are less intact, the cell has a higher tendency to lose cytoplasmatic contents as a reflection of the reduced cell organization [6,19,27]. The KL results for the first year of the evaluation showed the opposite effect to what would be expected, but this lower value observed for the T1 treatment is due to the influence of frequent fertilization with KCl before the evaluation. In the second year of evaluation, after the establishment of a new K dynamic in the soil and the reduction of Cl levels, less KL was found in treatments T4, T5, and the control.

Another piece of evidence for the reduction in the quality of the coffee beans and beverage with increasing doses of KCl is the high values of the EC observed in treatments T1, T2, and T3. As KL, CE also has a direct relationship with the integrity of the cell membrane [28,29].

Despite being considered indicatives of the quality of the beverage, these variables should not be decisive to vouch for the quality of the coffee [28]. In fact, the results of the cup quality in the last harvest suggest the same tendency observed for these variables. Notably, there was a response in the cup quality after the application of the treatments in the second harvest. As previously stated, in the first harvest, all response variables were very dependent on the previous fertilization in the area, thus the lack of response in the sensorial analysis.

However, from the second year of evaluation, some important facts should be emphasized about the K nutrition with the blends of fertilizers and the quality of the coffee beverage. Despite the lower scores for T1 and T2, the same behavior was observed for the control without K fertilization. This result suggests that only reducing the application of Cl via KCl is not enough to improve the quality of the beverage, but also maintaining adequate levels of K is essential to produce a high-quality coffee [6,15].

In the last harvest, treatment T5 achieved the highest score (89 points) in the sensorial analysis. T3 and T4, however, reached a few points less (86 points) than T5. Considering the higher cost of K_2_SO_4_ in relation to KCl, the choice for the composition of the K fertilizer should consider the economic cost that this difference of 3 points in the cup quality might return. Another important consideration is that there was a tendency for higher yield in T3 and T4 treatments despite the lack of statistical differences among the treatments. The difference between both treatments in relation to treatment T5 yielded more than five sacks of 60 kg of coffee beans, suggesting that yield should also be considered when choosing the best K fertilizer composition.

### 3.2. Principal Component Analyses for the Agronomic Variables and the Quality of the Coffee

The PCA results suggest that studies on how the fertilization of coffee plants affects the quality of the coffee beverage should be carried out for a long duration.

Overall, there were increased effects of the treatments after the second year of evaluation, probably due to the previous fertilizations with KCl, which is the most used source of K in Brazil [15]. However, some points should be considered in relation to the first harvest, such as the correlation among the variable Cl content in the beans, K content in the leaves, and yield showing the importance of the K fertilization in coffee plants. Nonetheless, the negative correlation of these variables with the cup quality suggests an unfavorable effect of one of these variables on the quality of the beverage, most probably the Cl content in the beans.

The correlation between EC and KL can be explained by the direct relationship shared by these two variables since they both indicate damages to the cell integrity of the beans [28,29,30]. The PCA confirms these results. These damages can lead to the loss of compounds related to the quality of the beans and the cup quality; therefore, lower EC and KL indicate lower coffee quality [28,31]. This lower bean quality is confirmed by the high negative correlation between PPO activity and the variables EC, PL, and TTA, showing that higher values of EC, KL, and TTA are associated with low PPO activity. Several studies found a positive correlation between the PPO activity and the sensorial quality of the coffee [32,33]. Thus, it is possible to conclude that there is a reduction in the PPO activity and the quality of the beverage as EC, KL, and TTA increase. In fact, damages to the cell membrane lead to the loss of selective permeability, facilitating the reaction of PPO with the phenolic compounds (the specific substrate of this enzyme). This reaction produces quinones that inhibit the activity of PPO [16,32].

Noteworthy, the high positive correlation between cup quality and the content of TS in the beans indicates a direct relationship where the increase in TS also increases the cup quality. Treatment T2 was close to this correlation, confirming the results for cup quality and indicating that increases in the K_2_SO_4_ proportion tend to increase cup quality. These results confirm the study of Silva et al. [16], which also involved doses and sources of K, and they add information about the sensorial quality of the coffee beverage. These findings provide evidence for the increase in the content of TS and better scores on the cup quality test as the proportion of K_2_SO_4_ in the blend also increases.

The reduction in the quality of the coffee beans is observed in the high correlation among KL, the stock of Cl in the soil, and the Cl content in the leaves. When the values of the variables related to Cl increase, K also increases, and the quality of the beans and the beverage reduces.

In the last harvest (2019/2020), a direct effect of the Cl in the variables related to the quality of the beverage is notable. Despite the fact that the PPO activity did not show differences for the treatments in both harvests, this variable behavior within the PCA is enough to indicate the direct relationship of this enzyme with the quality of the coffee beverage.

In addition, the high negative correlations of cup quality and PPO in relation to the variables related to Cl (stocks in the soil and content in the leaves) confirm the negative influence of the Cl in the beverage. In addition, treatment T5, which received only K_2_SO_4_, is closely related to cup quality and PPO activity.

Previous reports state that Cl increases the water content of the coffee fruits with consequent microbial fermentation [13,14]. We believe this explanation does not relate to this study since the beans were collected manually at the stage of cherry and benefited under controlled conditions, unlikely leading to an undesirable fermentation.

Although we did not perform physiological or morphological analyses of the coffee beans, the results allow us to infer that there is an effect of the Cl in the beans and that it might be related to the loss of quality of the beverage. The rationale is that the Cl can inhibit the activity of the PPO enzyme when reacting with the copper activator, thus reducing the enzymatic activity when KCl is applied [34].

In conclusion, this study showed that the blends of K fertilizers responded positively to the quality of the coffee beverage when the proportion of K_2_SO_4_ relative to KCl was increased. There was a tendency for higher KL, EC, and TTA with the increase of the KCl proportion, which might lead to damages in the cell membrane caused by the Cl and the consequent reduction of the PPO activity and quality of the beverage. However, the decision to use a determined blend of K should consider the improvement of the beverage and the economic return to the farm. Moreover, it should also consider the yield. For example, in this study, the blend that provided the best quality for the coffee beverage was not always the same responsible for the highest yield. And finally, it should also consider the economic costs of fertilization with K_2_SO_4_, which is more expensive than KCl. Considering these aspects, a management strategy could be the separation of the farm into plots based on the tendency to produce better quality coffees in previous years. In this case, each plot would receive a determined blend of K, and the highest proportions of K_2_SO_4_ should be applied to the plots with a tendency to produce higher quality coffees while the higher proportion of KCl would fertilize plots of low-quality coffee. Finally, any investigation with similar objectives to this work should perform a broader study, especially covering the regions of greater coffee production, where each plot/farm or region would receive potassium fertilization depending on the tendency for better or worse quality of the coffee beverage in the cup.

## 4. Materials and Methods

### 4.1. Experimental Area Characterization

The experiment was performed through three consecutive years (harvests of 2017/2018, 2018/2019, and 2019/2020) in a commercial production system of coffee located in the municipality of Santo Antônio do Amparo-MG, Brazil (20°53′26.04″ S and 44°52′04.14″ W and mean altitude of 1100 m). The plantation of Coffea arabica L., cultivar Catuaí Vermelho IAC 99, initiated in 2012 and spaced at 3.40 m × 0.65 m, is planted on a clayey Dystrophic Red Latosol-Latossolo Vermelho distrófico (Oxysol) [35].

Before the experiment, soil samples were collected for chemical attributes and texture analyses (Table 4). Samples from the 0–80 cm layer of soil were collected to assess K and Cl stocks. Undisturbed soil samples were taken to assess bulk density (BD). For depths over 5 cm, multiple samples were taken followed by the weighted average of the BD values. After determining K and Cl concentrations (mg kg^−1^), the values were multiplied by the BD to transform them into kg ha^−1^.

### 4.2. Experimental Design

The experimental design was in randomized blocks, with four blocks disposed at 90 degrees with the slope of the area. The treatments were composed of blends of KCl and K_2_SO_4_ (both in terms of K_2_O) as follows: T1—100% as KCl; T2—75% as KCl + 25% as K_2_SO_4_; T3—50% as KCl + 50% as K_2_SO_4_; T4 –25% as KCl + 75% as K_2_SO_4_; T5: 100% as K_2_SO_4_; and a control without K_2_O application. Each plot was composed of three planting lines with 16 plants, and the 10 central plants were considered a useful area (Figure 11).

### 4.3. Experiment Conducting

#### 4.3.1. Liming, Fertilization, and Gypsum Application

After the coffee harvest of each studied year, soil samples from the 0–10 cm, 0–20 cm, and 20–40 cm layers were collected to evaluate the needs for liming, fertilization, and gypsum application, respectively [17]. Liming was applied at 1.0 t ha^−1^, 1.2 t ha^−1^, and 1.5 t ha^−1^ on the first, second, and third harvest years, respectively. Gypsum was applied at 1.1 t ha^−1^ and 2.0 t ha^−1^ in the second and third years, respectively. Both dolomite lime and gypsum were applied underneath the projection of the tree canopies. P was applied at 120, 90, and 90 kg ha^−1^ of P_2_O_5_ as triple superphosphate on each consecutive year. N was applied at 350, 350, and 400 kg ha^−1^ of N as ammonium nitrate on each consecutive year, divided into three applications.

#### 4.3.2. Potassium Fertilization

Before the experiment, the saline index of each blend of K fertilizer was determined by comparing a 10 g L^−1^ sodium nitrate solution with solutions prepared with the blends of KCl e K_2_SO_4_ at the same concentration (Jackson 1958). The electric conductivity of the solutions and the saline index were calculated according to the equation: SI = [((ECa))⁄((ECb))] × 100, where SI is the saline index, ECa is the electric conductivity of the sample, and ECb is the electric conductivity of the sodium nitrate solution. The SI found were 142, 130, 121, 112, and 104%, for T1, T2, T3, T4, and T5, respectively.

The maintenance fertilization was done according to Guimarães et al. [36] using the abovementioned blends. The doses of K_2_O applied were 150, 200, and 300 kg ha^−1^ for the respective agricultural years of 2017/2018, 2018/2019, and 2019/2020. All K fertilizations were divided into three applications.

#### 4.3.3. Agronomic Variables Assessed on the Three Harvests

K and Cl content in the leaves

The third and fourth pair of leaves on both sides of the plants were collected from the useful area 20 days after the second application of the cover fertilization. The leaves were washed in deionized water, dried at 65 °C, and grounded in a Willey mill. The plant material was digested in a solution of nitric-perchloric acid (4 parts of nitric acid to 1 part of perchloric acid), and K was determined with inductively coupled plasma (ICP). To determine Cl, 1 g of grounded material was added to 50 mL of ultrapure water under agitation for 15 min [37]. After filtering the extract, the content of Cl (mg kg^−1^) was determined with a selective electrode (Hanna^®^, model HI4107) coupled to a Hanna^®^ device, model HI2221. The determination curve was built using the concentrations of 2, 20, 200, and 1000 mg L^−1^ of Cl.

Yield

The harvests were done when more than 70% of the fruits were mature. For the chemical analyses, 4 L of beans at the cherry stage were collected two days before each harvest. Fruits were peeled with an electric peeler (Pinhalense^®^, model DPM-02) and submerged for 24 h to remove the mucilage. After removing the peels and the rotten beans, samples were air-dried until a 10.8% to 11.2% moisture level.

The yield was determined by harvesting all fruits in the useful area. After the harvest, 5 L of a mix of fruits in every stage of maturation were air-dried under sunlight for one day. When the samples reached around 12% of moisture, beans were peeled and weighted. The moisture level was then adjusted to 12%, which is considered adequate for commercialization. To estimate yield, the weight of the beans in the useful area was projected to the number of plants in one hectare (4524 plants).

K and Cl content and removal in the beans

K and Cl contents in the beans were determined at the cherry stage after air-drying (65 °C, until constant weight) and grounding the beans in a Willey mill. K content was determined after nitric-perchloric digestion with measures done by ICP. Cl content followed the same procedures to quantify Cl in the leaves. The amounts of these elements removed from the soil were obtained by multiplying their content in the beans by the yield on each treatment.

Stocks of Cl in the soil

Stocks of Cl in the 0–20 and 20–80 cm layers of soil were checked during the experiment. Six soil samples were taken from the soil underneath the projection of the tree canopies (three from each side of the parcel). Extraction and determination of Cl followed the same procedure described for leaf Cl content, but with the proportion of 10 g of soil to 50 mL of ultrapure water. The stocks were determined by multiplying the element concentration by the mass of soil in each layer.

The analytical standard Tomato leaves (NIST 1573A), with 0.66% of Cl, was used in both soil and plant material analysis. The mean recovery of Cl was higher than 92%, assuring that the extraction and determination used for Cl were effective for both soil and plant material.

Chemical analysis of the beans and coffee sensorial analysis

After benefiting the coffee samples, the beans were stored in paper bags in a cold chamber until the chemical and sensorial analyses. The chemical analyses were performed at the Laboratory of Coffee Quality Analysis in the Empresa de Pesquisa Agropecuária de Minas Gerais (EPAMIG). Potassium leaching (KL, in µg g^−1^) was determined after 5 h of soaking [29], and electric conductivity (EC, in µS cm^−1^ g^−1^) was determined according to Loeffler et al. [38]. The total titratable acidity (TTA, m mL NaOH 0.1 N 100 g^−1^) was done according to Carvalho et al. [32] in the adaptation of the methodology from the Association of Official Analytical Chemists [39]. The content of total sugars (TS, in %) followed the anthrone method [40]. The activity of the polyphenol oxidase enzyme (PPO, in u min^−1^ g^−1^) was determined according to Carvalho et al. [32]. Total phenolic compounds (Pol, in %) were extracted according to Goldstein and Swain [41] and determined by the Folin-Denis method, described by AOAC [39]. Caffeine content (Caf, in %) was determined by spectrophotometry at 273 nm [42]. The coffee beans were frozen in liquid nitrogen and grounded in an IKA mill for the analyses, except for the KL and EC determinations.

The sensorial analysis (cup quality) was performed at the Laboratory of Agricultural Products Processing in the Universidade Federal de Lavras following the Specialty Coffee Association of America (SCAA) protocol. Three professionals with skills to differentiate fragrances, characteristics, and flavors participated in the cup test. The evaluation was based on scores given to the following attributes: fragrance/aroma, uniformity, clean cup, sweetness, flavor, acidity, body, aftertaste, balance, defects, and overall. The coffees were classified as the SCAA [43] according to their final scores (Table 5).

Statistical analyses

After model validation and analysis of variance indicating differences among treatments (*p* < 0.05), the response variables were submitted to Tukey’s test (*p* < 0.05) on the R 3.3.1 environment [44]. Principal component analyses (PCA) were performed to correlate the agronomic variables with the coffee beverage variables and yield. In the PCA, two components (Dim1 and Dim2) were used to represent the total data variability. The package Facto MineR (version 1.42) was used in the R software.

## 5. Conclusions

The activity of the polyphenol oxidase enzyme and the cup quality indicate that the ion Cl- reduces the quality of the coffee beverage. The increased application of the Cl- ion increases KL, EC, and TTA, indicators of the loss of coffee quality. K content in the leaves was not influenced by the application of blends of K fertilizer while Cl content increased linearly with KCl applied. The application of KCl and K_2_SO_4_ blends influenced coffee yield and the optimum proportion was 25% of KCl and 75% of K_2_SO_4_. The highest score in the cup quality test was observed with 100% K_2_SO_4_. However, other blends showed close scores. The decision for the fertilizer should consider the cost of the K source. KL and EC can indirectly show that the Cl can damage the coffee beans and reduce the selective permeability of the cell membrane, with possible negative consequences to the coffee beverage.

## Figures and Tables

**Figure 1 plants-12-00885-f001:**
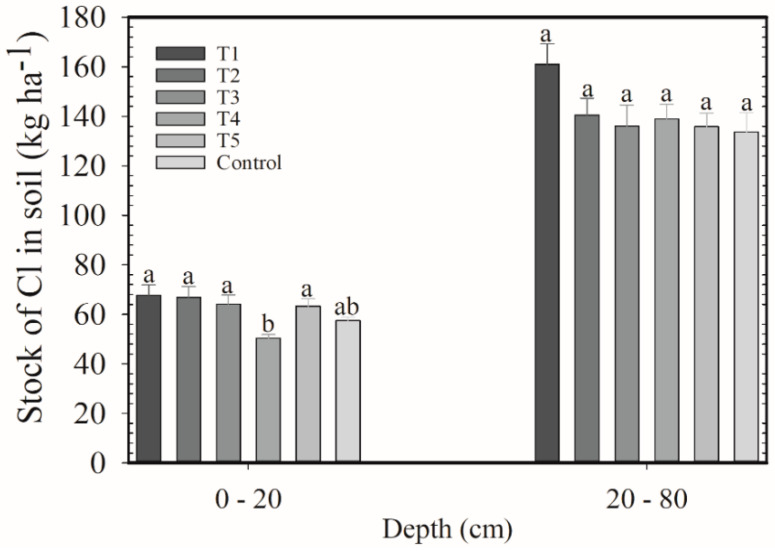
Stocks of Cl in the 0–20 and 20–80 cm layers after application of KCl and K_2_SO_4_ blends as cover fertilization on coffee plants. 2018/2019 harvest. Means followed by the same letter in the column do not differ according to Tukey’s test (*p* < 0.05). Vertical bars indicate the standard error of the mean (*n* = 4). T1: 100% KCl; T2: 75% KCl + 25% K_2_SO_4_; T3: 50% KCl + 50% K_2_SO_4_; T4: 25% KCl + 75% K_2_SO_4_; T5: 100% K_2_SO_4_; control did not receive K_2_O.

**Figure 2 plants-12-00885-f002:**
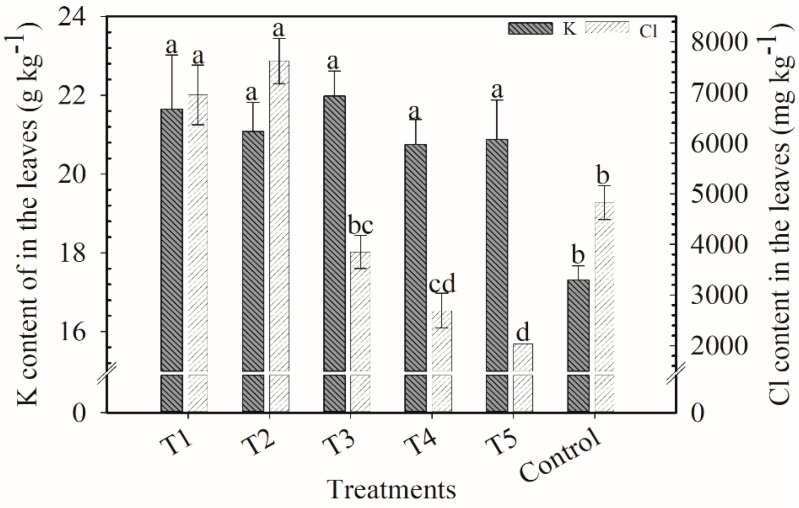
K and Cl contents in the leaves of coffee plants, 20 days after application of the second cover fertilization parcel. Harvest of 2018/2019. Means followed by the same letter in the column do not differ according to Tukey’s test (*p* < 0.05). Vertical bars indicate the standard error of the mean (*n* = 4). T1: 100% KCl; T2: 75% KCl + 25% K_2_SO_4_; T3: 50% KCl + 50% K_2_SO_4_; T4: 25% KCl + 75% K_2_SO_4_; T5: 100% K_2_SO_4_; control did not receive K_2_O.

**Figure 3 plants-12-00885-f003:**
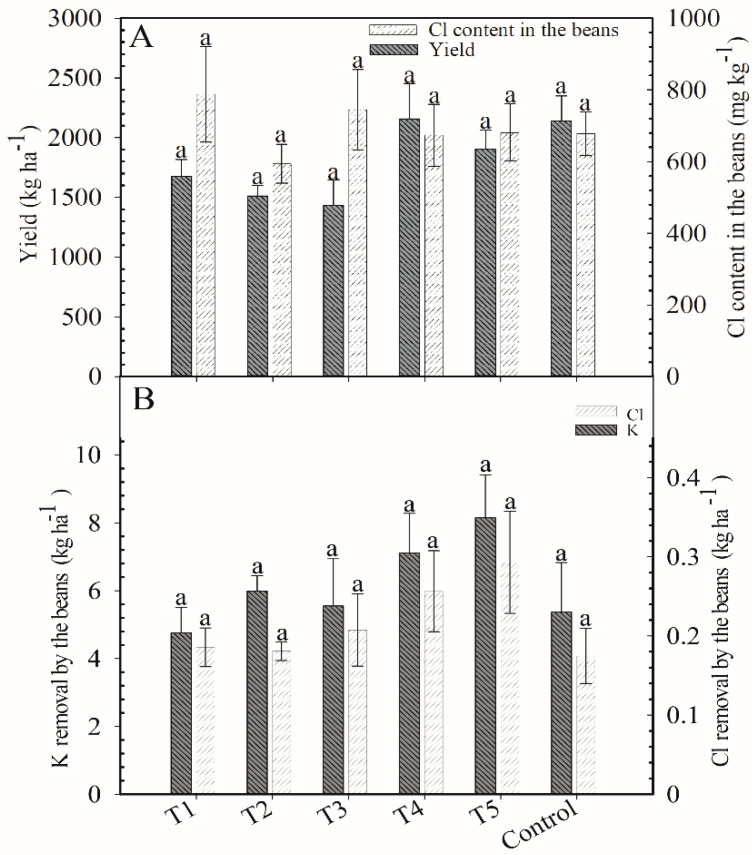
Yield of coffee plants and Cl content in the beans at cherry stage (**A**) and K and Cl removal by the beans (**B**) after application of blends of KCl e K_2_SO_4_ as cover fertilization. Harvest of 2018/2019. Means followed by the same letter in the column do not differ according to Tukey’s test (*p* < 0.05). Vertical bars indicate the standard error of the mean (*n* = 4). T1: 100% KCl; T2: 75% KCl + 25% K_2_SO_4_; T3: 50% KCl + 50% K_2_SO_4_; T4: 25% KCl + 75% K_2_SO_4_; T5: 100% K_2_SO_4_; control did not receive K_2_O.

**Figure 4 plants-12-00885-f004:**
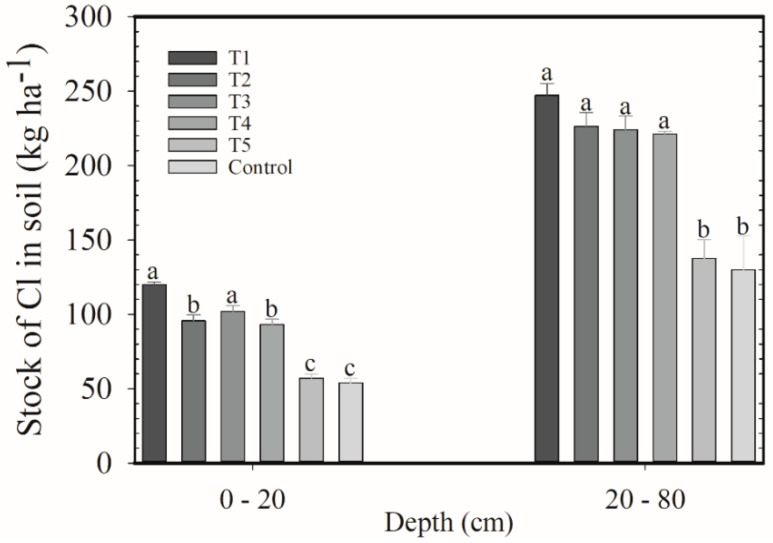
Stocks of Cl in the 0–20 and 20–80 cm layers after application of blends of KCl e K_2_SO_4_ as cover fertilization. Harvest of 2019/2020. Means followed by the same letter in the column do not differ according to Tukey’s test (*p* < 0.05). Vertical bars indicate the standard error of the mean (*n* = 4). T1: 100% KCl; T2: 75% KCl + 25% K_2_SO_4_; T3: 50% KCl + 50% K_2_SO_4_; T4: 25% KCl + 75% K_2_SO_4_; T5: 100% K_2_SO_4_; control did not receive K_2_O.

**Figure 5 plants-12-00885-f005:**
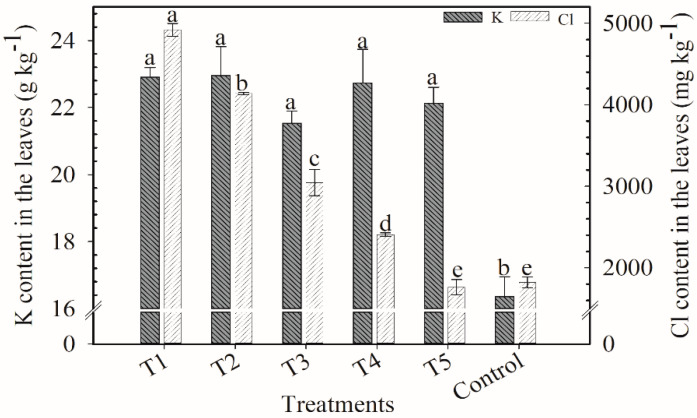
K and Cl content in the leaves of coffee plants. Harvest of 2019/2020. Means followed by the same letter in the column do not differ according to Tukey’s test (*p* < 0.05). Vertical bars indicate the standard error of the mean (*n* = 4). T1: 100% KCl; T2: 75% KCl + 25% K_2_SO_4_; T3: 50% KCl + 50% K_2_SO_4_; T4: 25% KCl + 75% K_2_SO_4_; T5: 100% K_2_SO_4_; control did not receive K_2_O.

**Figure 6 plants-12-00885-f006:**
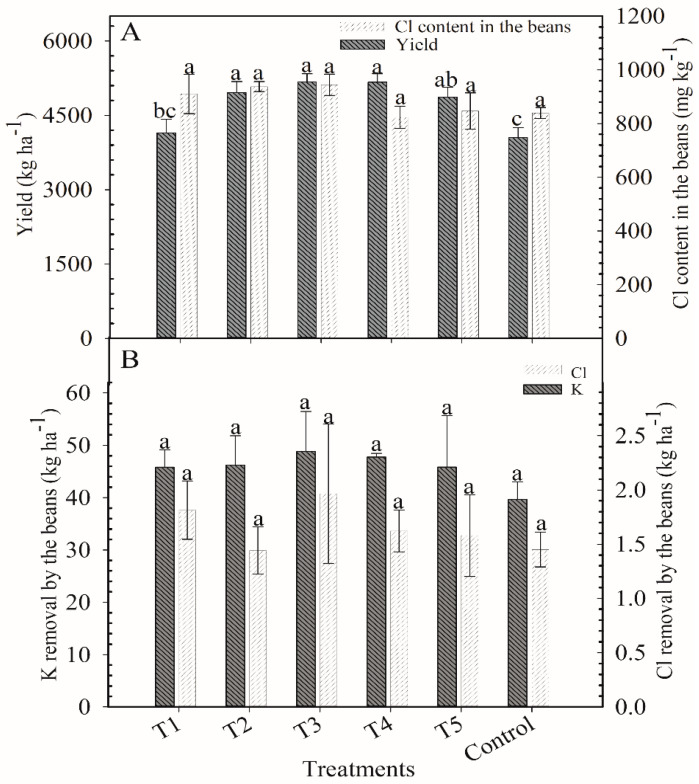
Yield (**A**), Cl content in the beans (**A**), and removal of K and Cl (**B**) by the beans of coffee at cherry stage after application of blends of KCl e K_2_SO_4_ as cover fertilization. Harvest of 2019/2020. Means followed by the same letter in the column do not differ according to Tukey’s test (*p* < 0.05). Vertical bars indicate the standard error of the mean (*n* = 4). T1: 100% KCl; T2: 75% KCl + 25% K_2_SO_4_; T3: 50% KCl + 50% K_2_SO_4_; T4: 25% KCl + 75% K_2_SO_4_; T5: 100% K_2_SO_4_; control did not receive K_2_O.

**Figure 7 plants-12-00885-f007:**
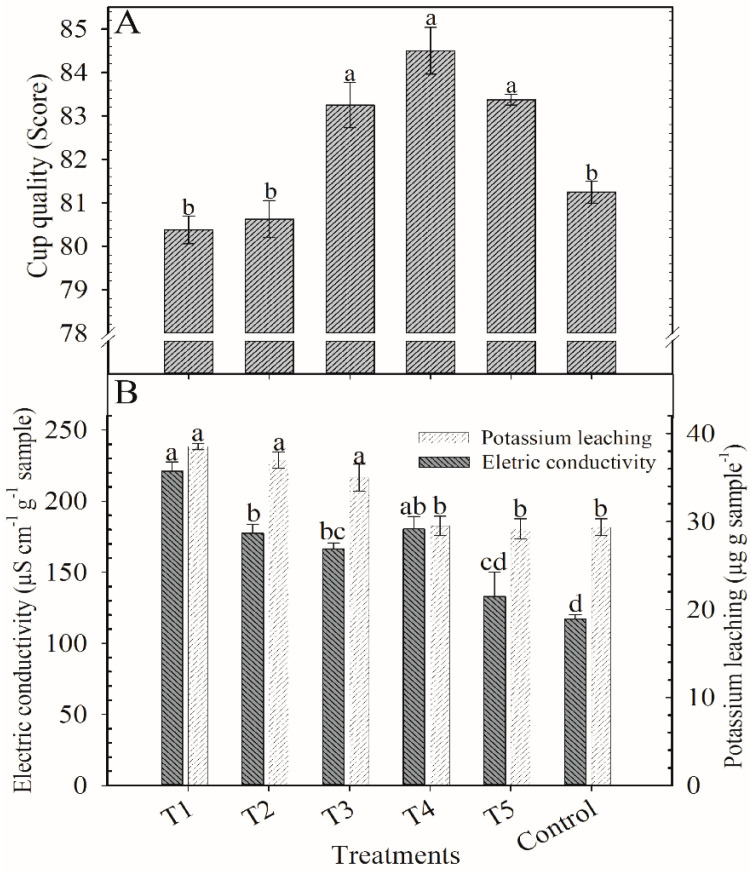
Scores of cup quality (**A**), electric conductivity and potassium leaching (**B**) in coffee beans at cherry stage after application of blends of KCl e K_2_SO_4_ as cover fertilization. Harvest of 2018/2019. Means followed by the same letter in the column do not differ according to Tukey’s test (*p* < 0.05). Vertical bars indicate the standard error of the mean (*n* = 4). T1: 100% KCl; T2: 75% KCl + 25% K_2_SO_4_; T3: 50% KCl + 50% K_2_SO_4_; T4: 25% KCl + 75% K_2_SO_4_; T5: 100% K_2_SO_4_; control did not receive K_2_O.

**Figure 8 plants-12-00885-f008:**
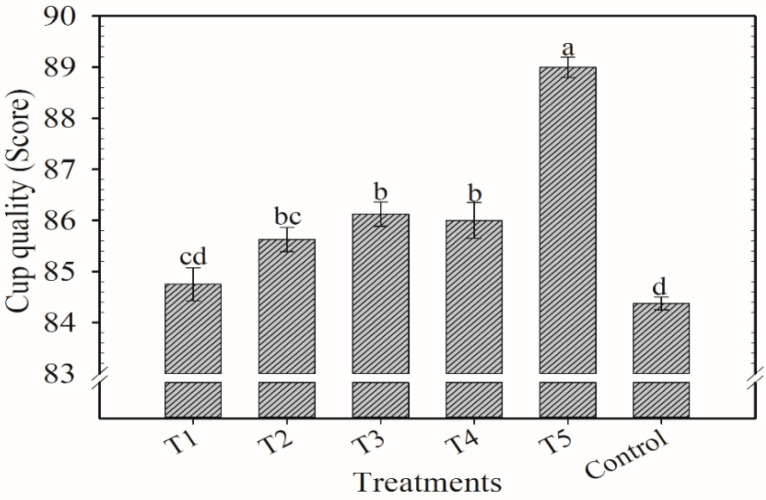
Scores of the cup quality of coffee beans at cherry stage after application of blends of KCl e K_2_SO_4_ as cover fertilization. Harvest of 2019/2020. Means followed by the same letter in the column do not differ according to Tukey’s test (*p* < 0.05). Vertical bars indicate the standard error of the mean (*n* = 4). T1: 100% KCl; T2: 75% KCl + 25% K_2_SO_4_; T3: 50% KCl + 50% K_2_SO_4_; T4: 25% KCl + 75% K_2_SO_4_; T5: 100% K_2_SO_4_; control did not receive K_2_O.

**Figure 9 plants-12-00885-f009:**
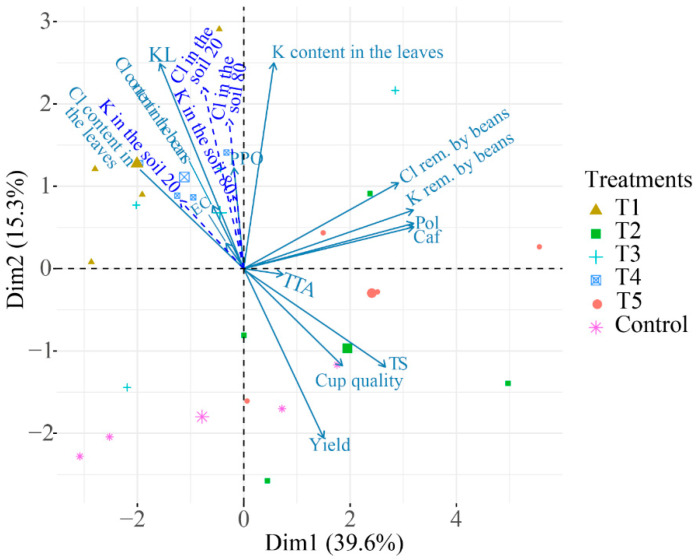
Principal component analysis for the harvest of 2018/2019. PPO = activity of the enzyme polyphenol oxidase; KL = K leaching; TTA = total titratable acidity; EC = electric conductivity; Pol = total phenolic compounds; Caf = content of caffeine; TS = content of total sugars; K in the soil 20 = stock of K in the 0–20 cm layer; K in the soil 80 = stock of K in the 20–80 cm layer; Cl in the soil 20 = stock of Cl in the 0–20 cm layer; Cl in the soil 80 = stock of Cl in the 20–80 cm layer, K rem. by beans: K removal by the beans, Cl rem. by beans: Cl removal by the beans. T1: 100% KCl, T2: 75% KCl + 25% K_2_SO_4_, T3: 50% KCl + 50% K_2_SO_4_, T4: 25% KCl + 75% K_2_SO_4_, T5: 100% K_2_SO_4_.

**Figure 10 plants-12-00885-f010:**
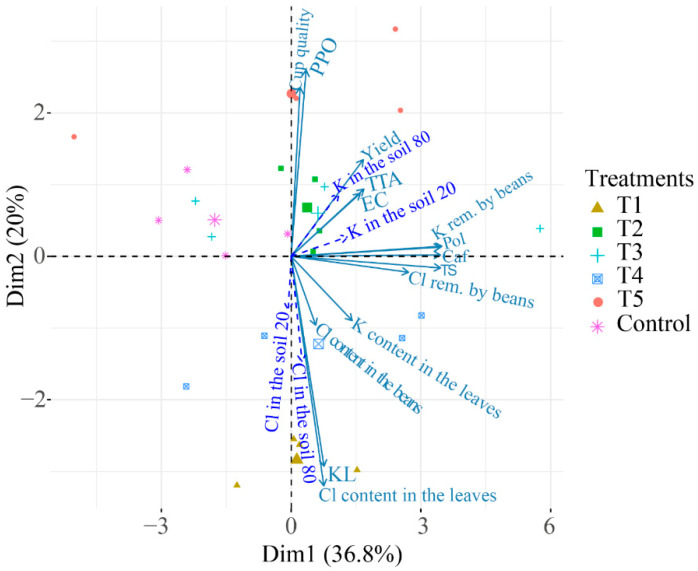
Principal component analysis for the harvest of 2019/2020. PPO = activity of the enzyme polyphenol oxidase; KL = K leaching; TTA = total titratable acidity; EC = electric conductivity; Pol = total phenolic compounds; Caf = content of caffeine; TS = content of total sugars; K in the soil 20 = stock of K in the 0–20 cm layer; K in the soil 80 = stock of K in the 20–80 cm layer; Cl in the soil 20 = stock of Cl in the 0–20 cm layer; Cl in the soil 80 = stock of Cl in the 20–80 cm layer, K rem. by beans: K removal by the beans, Cl rem. by beans: Cl removal by the beans. T1: 100% KCl, T2: 75% KCl + 25% K_2_SO_4_, T3: 50% KCl + 50% K_2_SO_4_, T4: 25% KCl + 75% K_2_SO_4_, T5: 100% K_2_SO_4_.

**Figure 11 plants-12-00885-f011:**
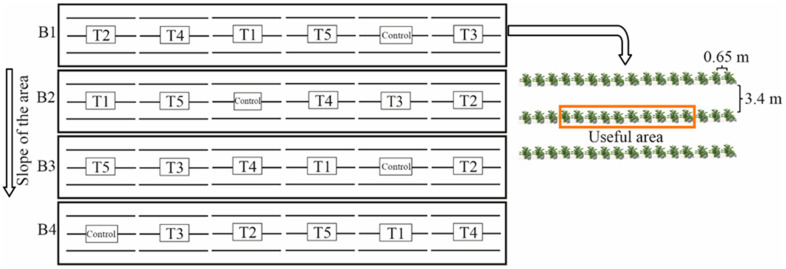
Schematic representation of the experimental design, number of plants in each plot, and the useful area used to collect the data.

**Table 1 plants-12-00885-t001:** Initial content and stocks of K and Cl in the soil and in the leaves of coffee plants.

Depth	BD	K^+^	Cl^−^	S.K^+^	S.Cl^−^	Leaf K^+^ Foliar	Leaf Cl^−^
cm	g cm^−3^	mg dm^−3^		kg ha^−1^		g kg^−1^	mg kg^−1^
0–20	1.1	91.5	153.8	201.3	338.3	19.2	2880
20–80	1.1	58.6	203.8	386.7	1345.0		

BD = Soil density by the volumetric ring method; S.K^+^ = stocks of K; S.Cl^−^ = stocks of Cl. Both stocks were calculated by multiplying the content of the element by the mass of soil in the layer.

**Table 2 plants-12-00885-t002:** Variables analyzed in the coffee beans for the harvest of 2018/2019.

Treatments	Pol	TS	Caf	PPO	TTA
T1	4.9a	9.4a	1.05a	47.0a	188.6a
T2	5.1a	9.0a	1.03a	45.7a	189.7a
T3	4.8a	8.9a	1.02a	47.8a	193.1a
T4	5.0a	9.5a	1.03a	45.9a	194.9a
T5	5.1a	9.0a	1.03a	47.7a	191.3a
Control	5.2a	8.9a	1.02a	44.5a	187.1a
CV (%)	6.9	5.3	2.8	8.5	2.9
Mean	5.0	9.1	1.03	46.4	190.8

CV (%) = coefficient of variation; Pol = total phenolic compounds (%); TS = content of total sugars (%); Caf = content of caffeine (%); PPO = polyphenol oxidase activity (u min^−1^ g^−1^); TTA = total tritable acidity (mL NaOH 0.1 N 100 g^−1^). Means followed by the same letter in column do not differ according to Tukey’s test (*p* < 0.05). T1: 100% KCl; T2: 75% KCl + 25% K_2_SO_4_; T3: 50% KCl + 50% K_2_SO_4_; T4: 25% KCl + 75% K_2_SO_4_; T5: 100% K_2_SO_4_; control did not receive K_2_O.

**Table 3 plants-12-00885-t003:** Variables analyzed in the coffee beans for the harvest of 2019/2020.

Treatments	Pol	TS	Caf	PPO	KL	TTA	EC
T1	5.1a	9.7a	1.05a	47.4a	72.7a	194.0a	119.4a
T2	5.2a	9.7a	1.09a	57.8a	69.9a	195.5a	129.7a
T3	5.1a	9.5a	1.08a	57.5a	62.8a	194.8a	119.7a
T4	5.1a	9.7a	1.06a	52.1a	75.2a	196.4a	126.7a
T5	5.2a	9.6a	1.03a	55.6a	80.2a	197.3a	134.7a
Control	5.0a	9.4a	1.03a	54.0a	64.2a	192.5a	118.4a
CV (%)	3.5	3.5	6.1	11.6	13.8	3.7	12
Mean	5.1	9.6	1.06	54.1	70.9	195	124

CV (%) = coefficient of variation; Pol = total phenolic compounds (%); TS = content of total sugars (%); Caf = content of caffeine (%); PPO = polyphenol oxidase activity (u min^−1^ g^−1^); TTA = total tritable acidity (mL NaOH 0.1 N 100 g^−1^). Means followed by the same letter in column do not differ according to Tukey’s test (*p* < 0.05). T1: 100% KCl; T2: 75% KCl + 25% K_2_SO_4_; T3: 50% KCl + 50% K_2_SO_4_; T4: 25% KCl + 75% K_2_SO_4_; T5: 100% K_2_SO_4_; control did not receive K_2_O.

**Table 4 plants-12-00885-t004:** Soil analyses results on September 2017.

**Depth**	**pH CaCl_2_**	**K^+^**	**P**	**Ca^2+^**	**Mg^2+^**	**Al^3+^**	**H + Al**	**BS**	**ECEC**	**CEC**	**V**	**m**
cm	-	mg dm^−3^		cmol_c_ dm^−3^							%	
0–10	5.2	96	7.9	1.4	0.7	0.3	8.7	2.5	2.8	11.2	22.2	11.9
10–20	5.2	87	9.2	1.5	1.1	0.1	6.7	3.0	3.1	9.7	30.8	5.6
20–40	5.1	69	7.1	1.6	1.0	0.1	6.5	3.0	3.1	9.5	31.5	5.0
**Depth**	**OM**	**P(rem)**	**Zn^2+^**	**Fe^2+^**	**Mn^2+^**	**Cu^2+^**	**B**	**S**	**Sand**	**Silt**	**Clay**	
cm	dag kg^−1^	mg L^−1^	mg dm^−3^						%			
0–10	3.8	17.2	1.8	57.6	8.4	2.1	0.2	180	22	14	64	
10–20	3.7	15.9	2.2	52.4	7.7	2.1	0.3	90	22	16	62	
20–40	3.6	14.6	1.3	39.3	4.2	2.0	0.3	48	22	18	60	

P, K^+^, Fe^2+^, Zn^2+^, Mn^2+^, Cu^2+^—Mehlich extractor. Ca^2+^, Mg^2+^, Al^3+^—1 mol L^−1^ KCl extractor. H^+^ + Al^3+^—SMP extractor. B—hot water extractor. S—monocalcium phosphate in acetic acid extractor. BS = exchangeable bases sum. ECEC = effective cation exchange capacity. CEC = cation exchange capacity at pH 7.0. V = base saturation. m = aluminum saturation. P-rem = remaining phosphorus. OM = organic matter (oxidation with Na_2_Cr_2_O_7_ 0.57 mol L^−1^ + H_2_SO_4_ 5 mol L^−1^).

**Table 5 plants-12-00885-t005:** Coffee beverage classification according to the cup quality.

Final Score	Special Description	Classification
95–100	Outstanding	Super premium specialty
94–90	Excepcional	Premium specialty
85–89	Excellent	Specialty
84–80	Very good	Specialty
75–79	Good	Good quality—normal
74–70	Weak	Medium quality

Source: Specialty Coffee Association of America (SCAA) (2009).

## Data Availability

Not applicable.

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
