# Peer review of "Chloride Applied via Fertilizer Affects Plant Nutrition and Coffee Quality"

_plants, 2023, doi:10.3390/plants12040885_

Round 1

Reviewer 1 Report

The manuscript present the results of a three-year long experiment with different blends of KCl and K2SO4 to supply potassium to coffee plants, and the effects of these blends.

I recommend this manuscript to be accepted after moderate changes.

The coffee yield and K and Cl removal were not evaluated in the first harvest (2017/2018)? Why is that?

The plot in Figure 5B should be presented in Figure 6A, as it was done in Figure 3. 

The PCA plots should present the ellipsoid of each K blend treatment, to simplify the interpretation of those plots.

Why did the authors choose to use the agronomic variables (K and Cl removal, yield, etc.) as supplementary variables? It is not clear in the text. The authors used the least relevant variables, related to beverage quality, as the main variables in the PCA plots, which did not present differences between treatments (Table 3).

Additionaly, why did the authors choose to perform PCA for the second and third harvests separately? I recommend the authors to try performing only one PCA analysis with all the data from all three cropping cycles. 

Further comments are in the attached file.

Author Response

Dear reviewer,

Please see comments in review report.

Reviewer 2 Report

An interesting experience from the scientific and practical point of view regarding the cultivation of the species under study. Well written introduction and interesting discussion, especially from the scientific angle. The material and methods did not include some important information that was suggested. The result section is hard to read due to the complicated description of the results. I suggest focusing more on the significance of the differences in the obtained results, and not on their range. Placing two not very accurate units on the scale of the figures makes them difficult to read, which is confusing. Perhaps the results in the tables would be clearer.

Author Response

(The authors gave the same response as above.)

Reviewer 3 Report

Title:

The main object of this research was not clear in the present title, “nutrition” was aimed plant or soil? so the title should be revised.

Abstract:

1)      “Results show a reduction of the stocks of Cl in the soil.” Did a reduction of the stocks of Cl in the soil happen with all the fertilization treatments?

2)      “The highest score in the cup quality test was observed with 100% K2SO4. However, other blends showed close scores.” Suggest delete this sentence because the last sentence supplied the optimum proportion of applied fertilizer.

3)      The trend of the yield, the nutritional state of coffee plants, as well as the chemical composition and quality of the coffee beverage should be added with the consecutive  fertilization from 2017 to 2020.

Results:

Suggest the data of 2017/2018, 2018/2019 and 2019/2020 for one index list together, easily find the change with the extend of fertilization time. In the present experiments, consecutive  fertilization is an important factor to the effect of fertilizer treatments on crop yield and quality.

Author Response

(The authors gave the same response as above.)

Round 2

Reviewer 1 Report

I recommend this manuscript to be accepted for publication after minor changes.

In the answer for my first review the authors presented two plots regarding the Principal Component Analysis, with the ellipses of each treatment, and mentioned that the ellipses did not help simplifying the interpretation.

By considering the plots with the ellipses, I think that not showing them is not honest with the reader, because by presenting only the centroids, and not the ellipses, might induce the readers to interpret that the treatments did influenced the variables considered, which is not true.

I still suggest the authors to consider presenting the PCA plots obtained by with all variables, including the agronomical ones such as K and Cl uptake.

Author Response

Report- Reviewer 1

Reviewer 3 Report

This manuscript has been revised well and almost meet the requirements of the journal. However, there are the minor problem as followed. Figure 5, 6, 7, 8 should supply the titles of X axis. Line 514 and 594, it suggests that “of soil” is added after “layers”.

Author Response

Report - Reviwere 3
